# Multicomponent Domino Cyclization of Ethyl Trifluoropyruvate with Methyl Ketones and Amino Alcohols as A New Way to γ-Lactam Annulated Oxazacycles

**DOI:** 10.3390/molecules28041983

**Published:** 2023-02-20

**Authors:** Marina V. Goryaeva, Olesya A. Fefelova, Yanina V. Burgart, Marina A. Ezhikova, Mikhail I. Kodess, Pavel A. Slepukhin, Vasily S. Gaviko, Victor I. Saloutin

**Affiliations:** 1Postovsky Institute of Organic Synthesis, Ural Branch of the Russian Academy of Science (IOS UB RAS), Ekaterinburg 620108, Russia; 2M.N. Mikheev Institute of Metal Physics, Ural Branch of the Russian Academy of Sciences (IMP UB RAS), Ekaterinburg 620108, Russia

**Keywords:** multicomponent domino cyclizations, ethyl trifluoropyruvate, methyl ketones, amino alcohols, γ-lactams, tetrahydropyrrolo[2,1-*b*]oxazolones, tetrahydropyrrolo[2,1-*b*]oxazinones

## Abstract

A new route to bicyclic γ-lactams was found, which was proposed as a three-component cyclization of ethyl trifluoropyruvate with methyl ketones and 1,2-, 1,3-amino alcohols. As a result, a series of trifluoromethyl-substituted tetrahydropyrrolo [2,1-*b*]oxazol-5-ones and tetrahydropyrrolo[2,1-*b*][1,3]oxazine-6-ones was synthesized, in which the substituent at the nodal carbon atom was varied. The introduction of a twofold excess of ethyl trifluoropyruvate in reactions with amino alcohols and acetone made it possible to obtain the same bicycles, but functionalized with a hydroxyester fragment, which are formed due to four-component interactions of the reagents. Transformations with 2-butanone and aminoethanol lead predominantly to similar bicycles, while an analogous reaction with aminopropanol gives *N*-hydroxypropyl-2,3-dihydropyrrol-5-one. Almost all bicycles are formed as two diastereomers, the structure of which was determined using ^1^H, ^19^F, ^13^C NMR spectroscopy, including two-dimensional experiments and XRD analysis. A domino mechanism for the formation of tetrahydropyrrolo[2,1-*b*]oxazacycles was proposed, which was confirmed by their stepwise synthesis through the preliminary preparation of the aldol and bis-aldol from ethyl trifluoropyruvate and methyl ketones.

## 1. Introduction

The γ-lactam (2-pyrrolidone) framework is of great importance for the creation of new heterocyclic compounds, since this motif is present in many natural bioactive molecules [1], including alkaloid cotinine (I) found in tobacco [2], lactacystin (II), obtained from a Streptomyces bacterial strain [3] and clausenamide (III) extracted from Clausena lansium [4] (Figure 1). Some synthetic pharmaceuticals also have a γ-lactam moiety, for example, a respiratory stimulant doxapram (IV) [5], ethosuximide (V) used to treat absence seizures [6], and the large family of racetams that have nootropic and/or anticonvulsant effects [7]. One of the representatives of this series, dimiracetam (tetrahydropyrrolo[1,2-*a*]imidazole-2,5-dione) (VI) [8] is a bicyclic derivative of γ-lactam. Moreover, γ-lactam analogues of penem (VII) [9] and penicillin acids (VIII) [10] with antibacterial properties have been synthesized.

One of the known bicyclic γ-lactams are Meyers’ lactams (tetrahydropyrrolo[2,1-*b*]oxazol-5-ones) (IX) [11] (Figure 1), which have a great synthetic potential for obtaining natural alkaloids due to the possibility of opening the oxazole ring [12]. For their preparation, a bielectrophile–binucleophile reaction of γ-keto acids with chiral amino alcohols, called Meyers’ lactamization, is used [13,14]. In addition, oxazolo-annulated γ-lactams can be obtained by cyclization of levulinic acid with *R*-phenylglycinol [15] or by the reaction of hydroxyl halogenoamides with Michael acceptors [16,17]. The latter method was also used for the synthesis of oxazine-annulated γ-lactams (tetrahydropyrrolo[2,1-*b*][1,3]oxazine-6-ones) [17]. Cyclization of ethyl 4-oxoalkanoates with amino alcohols is also a convenient protocol for the synthesis of this bicyclic system [18,19,20].

The preparation of fluorine-containing tetrahydropyrrolo[2,1-*b*]oxazol-5-ones is limited to a few examples. Pentafluoroethyl- and tri(di)fluoromethyl-substituted derivatives were prepared by addition of CF_3_CF_2_Li to the *N*-valinol imide at low temperatures [21], or acid-catalyzed cyclization of phenylalaninol with methyl 5,5,5-trifluoro-4-oxopentanoate [22] or ethyl 5,5-difluoro-4-oxopentanoate [23], respectively. Electrophilic fluorination of tetrahydropyrrolo[2,1*-b*]oxazolones via their enolation followed by the reaction with *N*-fluorobenzenesulfonimide at −70–(−78 °C) made it possible to synthesize such mono-fluorinated bicycles [24]. It is obvious that all these methods have strict restrictions on the reagents introduced, and some of them require special equipment. Information on fluorine-containing tetrahydropyrrolo[2,1-*b*][1,3]oxazine-6-ones was not found by us. Although the synthesis of fluoroorganic compounds is gaining more and more popularity [25,26], due to the unique properties [27,28,29] that fluorine atoms introduce to molecules [30], as a result, they have more prospects as biologically active substances.

Multicomponent synthesis is the most modern, simple and low-cost way to create new molecules from available starting reagents. Over the past 7 years, our group has been developing a new multicomponent approach that makes it possible to obtain various fluorine-containing heterocyclic compounds from commercially available polyfluoroalkyl-3-oxo esters, methyl ketones, and nucleophiles [31,32,33,34].

This protocol is based on the outstanding ability of the polyfluoracyl group of the oxoester to attach the activated methylene group of the ketone. We have recently used this method for the synthesis of pyrrolidones annulated with an imidazole or pyrimidine ring based on the three-component reaction of ethyl trifluoropyruvate and methyl ketones with ethylenediamine or 1,3-diaminopropane [35]. It was found that, in contrast to similar transformations of trifluoroacetoacetic ester, the use of an excess of ethyl trifluoropyruvate in the reaction with acetone and diamines under microwave irradiation leads to tricyclic products with two pyrrolidone fragments.

In this work, for the synthesis of γ-lactams annulated with oxazole or oxazine rings (tetrahydropyrrolo[2,1-*b*]oxazol-5-ones and tetrahydropyrrolo[2,1-*b*][1,3]oxazine-6-ones), multicomponent reactions of ethyl trifluoropyruvate **1** with methyl ketones **2** and 1,2-, 1,3-amino alcohols **3** were studied. In addition, their differences from the previously studied cyclizations with 1,2- and 1,3-diamines were found [35].

## 2. Results

We started our study with a three-component reaction of ethyl trifluoropyruvate **1** with acetone **2a** and 2-aminoethanol **3a** in 1,4-dioxane at an equimolar ratio of reagents at room temperature (Figure 1), since previously such conditions were optimal in similar syntheses with diamines [35]. It turned out that the reaction in 1,4-dioxane proceeds nonselectively and, in addition to a mixture of *cis*- and *trans*-diastereomers of the expected 6-hydroxy-7a-methyl-6-(trifluoromethyl)tetrahydropyrrolo[2,1-*b*]oxazol-5-one **4a**, a small amount of ethyl 3,3,3-trifluoro-2-hydroxy-2[(6-hydroxy-5-oxo-6-(trifluoromethyl)tetrahydropyrrolo[2,1-*b*]oxazol-7a-yl)methyl]propanoate **5** is also formed as two *trans,cis*- and *cis,cis*-diastereomers (Figure 1, Table 1, entry 1). The *cis*- and *trans*-configurations of diastereomeric bicycles **4** were determined relative to the OH-group and oxygen atom of the adjacent heterocycle. Note that the *trans,cis*- and *cis,cis*- diastereomers **5** have an additional stereocenter in the hydroxyester fragment.

The formation of these products was recorded during the analysis of the reaction mixture by ^19^F NMR spectroscopy and GC-MS. It should be noted that in the GC-MS analysis diastereomers of bicycles **4a** and **5** had the same peaks of molecular ions, but different retention times. The use of ^19^F NMR spectroscopy is very informative in such studies, since the starting pyruvate **1** (δ_F_ CF_3_ 81.08 ppm) and products based on it have different chemical shifts of the signals of CF_3_ groups.

The non-selective reaction of ethyl trifluoropyruvate **1** with acetone **2a** and aminoethanol **3a** in 1,4-dioxane prompted us to investigate this synthesis in various solvents and with different amounts of pyruvate **1**. However, varying solvents (toluene, THF, dichloroethane, ethanol, acetonitrile) at equimolar loadings of reagents invariably led to the formation of a mixture of products **4a** and **5**, while the number of heterocycles **5** in the reaction mixture even increased (Table 1, entries 1–6).

Since bicycle **5** is the result of a four-component cyclization due to the participation of two molecules of trifluoropyruvate **1**, it was logical to study these transformations with its twofold excess. Indeed, it turned out that the use of a twofold excess of pyruvate **1** increased the yield of compounds **5**, while bicycle **4a** was formed in a minimum amount of 2–6% (Table 1, entries 7–9). However, using an excess of pyruvate **1**, we detected the formation of aldol **6a** [36] and bis-aldol **7a** [35] by ^19^F NMR spectroscopy, which were isolated and characterized earlier. The highest yield of heterocycles **5** was achieved in THF at room temperature (Table 1, entry 9). Heating the reaction mixture in THF at 50 °C to speed up the process resulted in resinification and an increase in by-products (Table 1, entry 10).

Thus, as a result of varying the conditions, it was found that 1,4-dioxane is the most optimal solvent for the preparation of heterocycle **4a**, and THF for the synthesis of product **5**.

Furthermore, we carried out three-component equimolar reactions of ethyl trifluoropyruvate **1** with methyl ketone **2a–d** and 2-aminoethanol **3a** or 3-amino-1-propanol **3b** in 1,4-dioxane at room temperature. In this case, the introduction of aminopropanol **3b** in the reaction expands the scope of these three-component transformations, allowing the synthesis of oxazine derivatives. Variation of the methyl ketone component, which used not only acetone **2a**, but also 2-butanone **2b**, 2-hexanone **2c**, and acetophenone **2d**, makes it possible to change the substituent at the nodal carbon atom of the resulting bicycles. It was found that in reactions with alkyl methyl ketones **2a–c** in each case, a mixture of *cis*- and *trans*-diastereomers of pyrrolo[2,1-*b*][1,3]oxazol-5-ones **4a–c** or pyrrolo[2,1-*b*][1,3]oxazin-6-ones **8a–c** is formed (Figure 2). Notably, there is one more regularity: *trans*-diastereomers were prevailed in the formation of oxazole derivatives **4**, while *cis*-isomers were prevailed in the formation of oxazine bicycles **8**.

In contrast, similar cyclizations of ethyl trifluoropyruvate **1** with amino alcohols **3a,b** and acetophenone **2d** in each case, lead to one diastereomer **4d^c^** or **8d^t^**. The change and increase in stereoselectivity of this reaction may be due to the presence of a bulky phenyl substituent, which plays the role of a conformational anchor stabilizing the most favorable diastereomeric form.

We succeeded in isolating diastereomers **4a^t^**, **4b^t^**, **4c^c^**, **4d^c^**, **8b^t^**, **8c^c^**, **8d^t^** in pure form by column chromatography. Diastereomers **4a^c^**, **4b^c^**, **4c^t^**, **8a^c^**, **8a^t^**, **8b^c^**, **8c^t^**, **8d^t^** contain from 2 to 19% impurities of the second isomer, but we were able to record ^13^C NMR spectra for them, in which signals of only the main compound were accumulated. The isolation of bicycles **4a^t^**, **4b^c^** and **8a^c^**, **8a^t^**, **8b^c^**, **8b^t^** obtained from acetone **2a** and 2-butanone **2b** was complicated by side products **5**, **9**, **10**, **11**, the individual synthesis of which will be described below (Figure 3 and Figure 4). These compounds were formed in a small amount, but strongly interfered with the separation, while no such behavior was observed in the reactions with 2-hexanone **2c**. We were unable to isolate the bicycle **4a^c^** in its pure form, and despite several column chromatography, it still contained impurities of by-products **4a^t^** (2%) and **5** (20%). Difficulties in separating diastereomers are due to their similar physicochemical properties owing to structural similarity.

Further, reactions of a twofold excess of pyruvate **1** with acetone **2a** and amino alcohols **3a,b** in THF at room temperature were studied. The reaction of pyruvate **1** with acetone **2a** and aminoethanol **3a** leads to the formation of pyrrolo[2,1-*b*][1,3]oxazolone **5** functionalized with a 2-trifluoromethyl-2-hydroxypropanoate fragment (which can be called a heterocyclic aldol) as a mixture of two *trans,cis*- and *cis,cis*-diastereomers in a ratio of 60%:40% (Figure 3), which we managed to separate. Diastereomer **5^tc^** precipitated during the reaction, and diastereomer **5^cc^** was isolated from the reaction mixture by column chromatography. However, the yields of pure products are low, since fractions with an unseparated mixture of isomers remain.

The reaction of a twofold excess of pyruvate **1** with acetone **2a** and aminopropanol **3b** proceeds similarly and leads to the formation of a mixture of diastereomers of functionalized pyrrolo[2,1-*b*][1,3]oxazinone **9** in a ratio of ~ 1:1 (Figure 3). However, due to their very similar properties, diastereomers **9** and **9’** were not separated.

In addition, 2-butanone **2b** was introduced into interaction with a twofold excess of pyruvate **1** and amino alcohols **3a,b** (Figure 4), and unexpected results were obtained. It turned out that the use of 2-butanone **2b** with pyruvate **1** and aminoethanol **3a** leads to the formation of 6-methyl-substituted pyrrolo[2,1-*b*][1,3]oxazolone as a mixture of diastereomers **10** and **10′** in a ratio of 56%:44% by analogy with the formation of bicycles **5** and **9** (Figure 3). In addition, *N*-hydroxyethylpyrrol-5-one **11a** was isolated from this reaction in a small amount.

The reaction of 2-butanone **2b** with pyruvate **1** and aminopropanol **3b** leads only to *N*-hydroxypropylpyrrol-5-one **11b** as a mixture of diastereomers **11b**:**11b’** in a ratio of 72%:28% (Figure 4). We were able to isolate diastereomer **11b** in pure form by fractional crystallization from a mixture of diethyl ether and hexane in 47% yield.

It is obvious that amino alcohols **3a,b** react as mononucleophiles during the formation of pyrrolidinones **11a,b**. Previously, we showed that amino alcohols **3a,b** in three-component reactions of polyfluoroalkyl-3-oxo esters with methyl ketones **2** or cycloketones can behave both as mono- and di-nucleophiles [32,37].

The introduction of 2-hexanone **2c** into the reaction with amino alcohols **3a,b** and a double excess of pyruvate **1** in THF led to the formation of bicycles **4c** and **8c** already obtained as a mixture of diastereomers (Figure 5). Obviously, the nucleophilicity of a-methylene center of the butyl substituent in 2-hexanone **2c** is significantly reduced under the influence of electronic and steric factors than in 2-butanone **2b** and, therefore, does not take part in the aldol addition reaction. It can be noted that these reactions were accompanied by the formation of more by-products, compared with the reactions performed at an equimolar ratio of reagents.

We were unable to select conditions for the synthesis of pyrrolo[2,1-*b*]oxazolones **4a,b** or pyrrolo[2,1-*b*]oxazinones **8a,b** in good yields in three-component reactions of trifluoropyruvate **1** with methyl ketones **2a,b** and amino alcohols **3a,b** due to the formation of side bicycles **5**, **9**, **10**, **11**, formed as a result of four-component transformations. In this regard, we used a two-stage approach through the initial preparation of aldols **6a,b** from ethyl trifluoropyruvate **1** and methyl ketones **2a,b** (Figure 6), thus, aldol **6a** was synthesized earlier [36], and the ethyl-substituted analog **6b** was obtained for the first time. Next, aldols **6a,b** were introduced into cyclization with amino alcohols **3a,b**, as a result of which bicycles **4a,b** and **8a,b** were also obtained as a mixture of *cis*- and *trans*-diastereomers.

We analyzed the reaction mixtures obtained by two- and three-component approaches using ^19^F NMR spectroscopy. A difference in the ratio of *cis*- and *trans*-diastereomeric products **4**, **8** (Table 2) was found, since the proportion of predominant isomers increased significantly. Thus, in the reactions of aldols **6a,b** with aminoethanol **3a**, *trans*-isomers **4a^t^**, **4b^t^** were formed with approximately a threefold advantage, and in reactions with aminopropanol **3b**, *cis*-diastereomers **8a^c^**, **8b^c^** increased by approximately two times. This made it possible to isolate diastereomers **4a,b^t^** and **8a,b^c^** in higher yields.

Heterocyclic aldols **5**, **9–11**, which are products of a four-component reaction, can be assumed to form in two ways: through the cyclization of bis-aldol **7** with amino alcohol **3** or through the addition of a methyl substituent of bicycles **4** and **8** to the trifluoroacyl group of pyruvate **1**. However, an attempt to carry out the aldolization reaction of pyruvate **1** under the action of the bicycle **4a^t^** was unsuccessful regardless of the conditions used (Figure 7). While the bis-aldol **7a** easily cyclized with aminoethanol **3a**, giving a mixture of diastereomers of the expected heterocyclic aldols **5^tc^** and **5^cc^** with a predominance of the *trans,cis*-form.

The structure of the synthesized heterocycles **4**, **5**, **8–11** was confirmed by IR, ^1^H, ^19^F, ^13^C NMR spectroscopy and mass spectrometry. The diastereomeric structure of bicycles **4**, **5**, **8**, **11** was established using two-dimensional experiments 2D ^1^H-^13^C HSQC, ^1^H-^13^C HMBC and X-ray diffraction analysis for **4d^c^**, **5**, **8c^c^**, **11b**. All diastereomers are racemates.

The synthesized bicycles **4a–d** and **8a–d** contain two asymmetric centers C-6(7) and C-7a(8a) (Figure 2). Analyzing the chemical shifts of the diastereotopic protons H-A and H-B in the ^1^H NMR spectra at C-7 or C-8 in heterocycles **4a–d**, **8a–d**, we found the following regularity: the values Δ_AB_ = δ_A_ − δ_B_ for the alkyl-substituted heterocycles **4a–c^c^** and **8a–c^c^**, which have the *cis*-configuration, are in the range Δ_AB_ 0.41–0.56 ppm, whereas for the trans-isomers **4a–c^t^** and **8a–c^t^** these values are much lower, Δ_AB_ 0.04–0.23 ppm. For diastereomers containing a phenyl substituent, the opposite pattern is observed, for example, for the *cis*-isomer **4d^c^** Δ_AB_ 0.10 ppm, while for the *trans*-diastereomer **8d^t^** Δ_AB_ 0.42 ppm (Table 3). It was found that the geminal spin–spin coupling constant of these protons of the *cis*-isomers **4a–d^c^**, **8a–c^c^** are ^2^*J* 15.2–15.5 Hz, while for the *trans*-diastereomers **4a–c^t^**, **8a–d^t^**
^2^*J* 14.2–14.9 Hz. Previously, we revealed similar features for the *trans*/*cis*-diastereomers of hexahydropyrrolo[1,2-*a*]imidazol-5-ones and hexahydropyrrolo[1,2-*a*]pyrimidin-6-ones [35].

Some regularities were found in the shifts of the signals of trifluoromethyl group in the ^19^F NMR spectra of pyrrolo[2,1-*b*]oxazolones **4a–d** and pyrrolo[2,1-*b*]oxazinones **8a–d**. Thus, the signals of the trifluoromethyl group of *cis*-isomers **4a–d^c^** are observed in the range δ_F_ 83.63–83.79 ppm, and the *trans*-forms **4a–c^t^** in a lower field δ_F_ 83.87–83.91 ppm, similarly for **8a–c^c^** signals are recorded in the region δ_F_ 83.83–83.88 ppm, and for **8a–d^t^** at δ_F_ 84.13–84.31 ppm.

The stereo configuration of tetrahydropyrrolo[2,1-*b*][1,3]oxazol-5-one **4** and tetrahydro-2*H*-pyrrolo[2,1-*b*][1,3]oxazine-6-one **8** was additionally confirmed by XRD analysis, which was performed for bicycles **4d^c^** (Figure 3a) and **8c^c^** (Figure 3b). It was found that these bicycles have the *cis*-arrangement of the OH-group in the pyrrole and the oxygen atom in the adjacent oxazacycle (Figure 3).

A more difficult task was to determine the structure of heterocyclic aldols **5^tc^**, **5^cc^**, **9**, **9’** and **10**, **10’**, which have three or four asymmetric centers, respectively.

The diastereomeric structure of pyrrolo[2,1-*b*]oxazol-5-one **5^tc^** was determined using XRD analysis (Figure 4a), conforming to which this compound is a racemic mixture of molecules having the configuration of substituents at the stereocenters C-3–*R**, C-5–*S**, C-9–*R** according to the numbering presented in Figure 4a. It was found that the hydroxyl substituent in the pyrrolidine ring and the oxygen atom of the oxazole backbone are in the *trans*-position, while this atom and the hydroxy group of the propanoate fragment are in the *cis*-position. The crystal packing of compound **5^tc^** is formed due to intermolecular hydrogen bonds of the lactam carbonyl and hydroxyl groups of the cycle O-2–H-2…O-1 1.816 Å and the ester carbonyl and hydroxyl group of the propanoate substituent O-4–H-4…O-5 2.054 Å (Figure 4b).

For a pair of isolated diastereomers **5^tc^**, **5^cc^** (Figure 2), two-dimensional experiments 2D ^1^H-^13^C HSQC and HMBC were performed, on the basis of which a complete assignment of signals in the ^1^H and ^13^C NMR spectra was made. The ^1^H NMR spectrum of pyrrolo[2,1-*b*]oxazol-5-one **5^tc^** is characterized by the presence of doublet signals of the methylene protons of the pyrrolidine ring H-7’’A and H-7’’B at δ_H_ 2.99, 2.29 ppm (Δ_AB_ 0.70 ppm, ^2^*J* 14.9 Hz) and propanoate substituent H-1’A and H-1’B at δ_H_ 2.63, 2.20 ppm (Δ_AB_ 0.43 ppm, *J* 14.0 Hz). The ^13^C NMR spectrum contains characteristic signals of carbonyl atoms at C-5’’ of lactam (δ_C_ 170.2 ppm) and at C-1 of ester (δ_C_ 167.8 ppm) fragments. The ^19^F NMR spectrum contains two singlet signals of trifluoromethyl groups at δ_F_ 83.83 and 84.92 ppm.

According to ^1^H, ^19^F and ^13^C NMR spectra, diastereomer **5^cc^** has a similar set of characteristic signals. However, analyzing the ^1^H NMR spectrum, it was found that the doublet signals of the methylene protons of the cycle at H-7’’A and H-7’’B (δ_H_ 2.61, 2.26 ppm) have a lower value Δ_AB_ 0.35 ppm and a smaller *J* constant of 14.0 Hz compared to the analogous values of the **5^tc^** heterocycle. It is obvious that for diastereomers **5^tc^**, **5^cc^**, containing a hydroxypropanoate fragment, the same trend in changing Δ_AB_ and constant *J* is observed, as for bicycle **4d^c^**, which has a bulky phenyl substituent. For the methylene protons H-1’A and H-1’B of the propanoate residue, resonating as doublet signals at δ_H_ 2.67, 2.50 ppm, the value of Δ_AB_ 0.17 ppm and *J* 15.0 Hz also changes. In the ^13^C NMR spectra of the **5^tc^** and **5^cc^** isomers, the largest differences in shifts were recorded for the carbon atoms C-7’’ (δ_C_ 42.1, 44.6 ppm) of the pyrrolidine ring and C-1 (δ_C_ 37.8, 40.8 ppm) of the propanoate substituent, which are adjacent to the C-7’’ and C-2 stereocenters (Figure 2). All these data allow us to suggest that the **5^cc^** bicycle has a *cis,cis*-diastereomeric structure, in which the position of the substituents at the C-7’’ and C-2 stereocenters changes compared to the **5^tc^** isomer.

The structure of pyrrolo[2,1-*b*][1,3]oxazin-6-ones **9**, **9’** and pyrrolo[2,1-*b*]oxazol-5-ones **10**, **10’** was also established using ^1^H, ^13^C and ^19^F NMR spectra, which contained a double set of all signals, since we were unable to separate diastereomers of bicycles **9** and **10**. However, their spectra characteristics were similar to those of bicycles **5^tc^** and **5^cc^**, that allowed us to assign them a similar structure, but without determining the diastereomeric structure due to close values of the chemical shifts of protons and carbon atoms in the ^1^H and ^13^C NMR spectra (see the experimental part).

To establish the diastereomeric structure of dihydropyrrol-5-ones **11a,b**, which have two asymmetric centers C-2 and C-4’’ (Figure 5), we used the data of ^1^H, ^13^C NMR spectroscopy and XRD analysis performed for **11b**. For compounds **11a** and **11b**, two-dimensional 2D ^1^H-^13^C HSQC and HMBC experiments were carried out, on the basis of which a complete assignment of signals in the ^1^H and ^13^C NMR spectra was made.

The diastereomeric structure of dihydropyrrol-5-one **11b** was determined by X-ray diffraction data (Figure 6a). Crystal packing is formed of a racemic mixture of molecules linked by intermolecular hydrogen bonds O-1–H-1…O-2 1.882 Å, O-3–H-3…O-6 2.122 Å (Figure 6b). The configuration of the substituents in the pyrrole ring at the C-1 stereocenter is *R**, and that of the propanoate substituent at C-8 is *S** (numbering is used according to X-ray diffraction data, Figure 6).

The ^13^C NMR spectra analysis of dihydropyrrol-5-ones **11b** and **11b’** revealed the presence of two downfield signals at δ_C_ 113.0–113.5 ppm and δ_C_ 137.1–138.4 ppm, which correspond to two sp^2^-hybridized carbon atoms C-3’’ and C-2’’, respectively.

In the ^19^F NMR spectra of diastereomers **11b** and **11b’**, the signals of the trifluoromethyl group of the pyrrole cycle (δ_F_ 85.74, 85.81 ppm) and the propanoate substituent (δ_F_ 85.81, 86.01 ppm) are observed in approximately the same range. However, according to the ^1^H NMR spectra, the nature of the signals of the methylene protons H-1’ of the propanoate fragment of isomers **11b** and **11b’** differs, which may indicate a different configuration of substituents at the adjacent C-2 stereocenter. Thus, the protons H-1’A and H-1’B in the spectrum of isomer **11b** resonate as two doublets at δ_H_ 3.19 and 2.97 ppm (Δ_AB_ 0.22 ppm, *J* 14.9 Hz), while the signals of the same protons of isomer **11b’** are observed as an AB system at δ_H_ 3.08 ppm (*J*_AB_ 16.7 Hz, Δ_AB_ 0.1 ppm). Taking into account that, according to X-ray diffraction analysis (Figure 6), the substituents at the C-2 stereocenter in diastereomer **11b** have the *S**-configuration. The difference in the nature of the resonation of the protons of the neighboring methylene group C-1’ allows us to assume the opposite *R**-configuration for the **11b’** isomer (Figure 5).

The ^13^C NMR spectrum of pyrrolone **11a** also contained characteristic low-field signals C-3’’ (δ_C_ 113.37 ppm) and C-2’’ (δ_C_ 137.90 ppm), confirming the presence of a double bond in the molecule. In its ^1^H NMR spectrum, methylene protons H-1’ resonate as a singlet at δ_H_ 3.16 ppm, which can be a degenerate AB system with Δ_AB_ 0 ppm, which is closer in nature to the signals of similar protons of isomer **11b’** (AB-system at δ 3.08 ppm, Δ_AB_ 0.1 ppm). Based on this, we assumed that compound **11a** has the *R**-configuration of substituents at C-2 (Figure 5).

Considering the mechanism of formation of bicycles **4a–d** and **8a–d** from ethyl trifluoropyruvate **1**, methyl ketones **2a–d** and amino alcohols **3a,b**, it can be safely assumed that three-component cyclizations are a sequential domino process (Figure 8). The first stage of which is aldolization, since by optimizing the conditions for the reaction of pyruvate **1** with acetone **2a** and aminoethanol **3a** (Figure 1, Table 1), we detected aldol **6a**.

In addition, we also experimentally demonstrated the feasibility of cyclization of aldols **6a,b** with amino alcohols **3a,b** into bicycles **4a,b** and **8a,b** (Figure 7), presumably proceeding through the condensation of the keto group of aldol **6** with the amino group of amino alcohol **3** leading to intermediate **X1**, which after tautomerization undergoes intramolecular cyclization involving ester and amino groups, providing dihydropyrrol-5-one **X2**. At the last stage, the formation of the second cycle occurs due to the intramolecular addition of a hydroxyl group to the double bond.

Similar processes can be assumed for the four-component formation of heterocyclic aldols **5** and **9**, only the stage of formation of bis-aldol **7** is added, which then cyclizes with amino alcohol **3** (Figure 8), forming dihydropyrrol-5-one **X4**. Such compounds were isolated and characterized in the case of Me-substituted derivatives **11a,b**. Subsequent intramolecular cyclization of pyrrolones **X4** gives bicyclic products **5**, **9**, **10**.

## 3. Material and Methods

### 3.1. Material

The solvents (acetonitrile, chloroform, hexane, diethyl ether and acetone **2a**) were obtained from AO “VEKTON” (St. Petersburg, Russia). 2-Butanone **2b**, 2-hexanone **2c**, 3-amino-1-propanol **3b** were purchased from Merck KGaA (Darmstadt, Germany). 2-Aminoethanol **3a**, acetophenone **2d** and 1,4-dioxane were obtained from Alfa Aesar (UK). Ethyl trifluoropyruvate **1** was purchased from ABCR (GmbH, Karlsruhe, Germany). The deuterosolvent DMSO was acquired from «SOLVEX» Limited Liability Company (Skolkovo Innovation Center, Moscow, Russia).

### 3.2. Methods

Melting points were measured in the open capillaries with a Stuart SMP3 melting-point apparatus (Bibby Scientific Limited, Staffordshire, UK). Two FT-IR spectrometer (Perkin-Elmer, Waltham, MA, USA) using the frustrated total internal reflection accessory with a diamond crystal. The ^1^H and ^19^F NMR spectra were registered on a Bruker DRX-400 spectrometer (400 or 376 MHz, respectively) or a Bruker AvanceIII 500 spectrometer (500 or 470 MHz, respectively) (Bruker, Karlsruhe, Germany). The ^13^C NMR spectra were recorded on a Bruker AvanceIII 500 spectrometer (125 MHz). The internal standard was SiMe4 (for ^1^H and ^13^C NMR spectra) and C_6_F_6_). The ^13^C chemical shifts were calibrated using the solvent signal DMSO-d_6_ (δ_C_ 39.5 ppm). For compounds **4a-d**, **8a-d**, **5**, **5’**, **11a**, **11b** signals in ^1^H and ^13^C spectra were assigned based on 2D ^1^H-^13^C HSQC and HMBC experiments. The high-resolution mass spectra (HRMS) were recorded on a Bruker maXis impact mass spectrometer (ESI) (Bruker, Karlsruhe, Germany). The column chromatography was performed on silica gel 60 (0.062–0.2 mm) (Macherey-Nagel GmbH & Co KG, Duren, Germany). The initial ethyl-2-hydroxy-4-methyl-4-oxo-2-(trifluoromethyl)butanoate (aldol **6a**) [35,36] and diethyl 2,6-dihydroxy-4-oxo-2,6-bis(trifluoromethyl)heptanedioate (bis aldol **7a**) [35] were synthesized by referring previously published methods.

### 3.3. General Procedures

*Synthesis of compounds **4** and **8** (**method A**)*: A solution of ethyl trifluoropyruvate **1** 1530 mg (9 mmol) and methyl ketone **2a–d** (9 mmol) in 1,4-dioxane (5 mL) was placed in a flat-bottomed flask. Then, amino alcohol **3a,b** (9 mmol) was added. The reaction mixture was stirred for 3–7 days at room temperature (25°C). After completion of the reaction (TLC and NMR ^19^F monitoring), the reaction mixture was concentrated on a rotary evaporator. The residue was triturated with hexane, and the resulting precipitate was collected by filtration and purified by recrystallization from an appropriate solvent (MeCN, Et_2_O), or by column chromatography (eluent: CHCl_3_, CHCl_3_–Et_2_O/1:1).

*Synthesis of compounds **5, 9, 10** and **11** (**method B**)*: A solution of ethyl trifluoropyruvate **1** 3060 mg (18 mmol) and methyl ketone **2a,b** (9 mmol) in THF (5 mL) was placed in a flat-bottomed flask. Then amino alcohol **3a,b** (9 mmol) was added. The reaction mixture was stirred at room temperature (25℃) for 4–7 days. After completion of the reaction (TLC and ^19^F NMR monitoring), the reaction mixture was concentrated on a rotary evaporator, the residue was purified by column chromatography (eluent: CHCl_3_–Et_2_O / 2:1, CHCl_3_–Et_2_O / 4:1). Product **5^tc^** precipitated out during the reaction. The precipitate was filtered and purified by recrystallization from MeCN. The filtrate was evaporated, purified by column chromatography (eluent CHCl_3_–Et_2_O/2:1), product **5^cc^** was obtained. Product **11b** was isolated from a mixture of diastereomers by fractional crystallization (hexane–diethyl ether / 1:3).

*Synthesis of products **4** and **8** (**method C**)*: A solution of aldol **6a,b** (5 mmol) in 1,4-dioxane (3 mL) was placed in a flat-bottomed flask. Then the amino alcohol **3a,b** (5 mmol) was added. The reaction mixture was stirred at room temperature (25 °C) for 4–5 days. After completion of the reaction (TLC and ^19^F NMR monitoring), the reaction mixture was concentrated on a rotary evaporator, the residue was purified by column chromatography (eluent: CHCl_3_–Et_2_O/1:1).

*Synthesis of compounds **5** (**method D**)*: A solution of bis-aldol **7a** (1990 mg, 5 mmol) in THF (3 mL) was placed in a flat-bottomed flask. Then the amino alcohol **3a** (305 mg, 5 mmol) was added. The reaction mixture was stirred at room temperature (25 °C) for 4–5 days. After completion of the reaction (TLC and 19F NMR monitoring), the reaction mixture was concentrated on a rotary evaporator, the residue was purified by column chromatography (eluent: CHCl_3_–Et_2_O / 1:1). Product **5^tc^** precipitated out during the reaction. The precipitate was filtered and purified by recrystallization from MeCN. The filtrate was evaporated, purified by column chromatography (eluent CHCl_3_–Et_2_O/2:1), product **5^cc^** was obtained.

### 3.4. Spectral Data

*(6R*,7aR*)-6-hydroxy-7a-methyl-6-(trifluoromethyl)tetrahydropyrrolo[2,1-b][1,3]oxazol-5(6H)-one (**4a^t^**).* Yield 36 % (729 mg, *method A*), 77% (886 mg, *method C*); white solid; m.p. 80°C (CHCl_3_–Et_2_O / 1:1). ^1^H NMR (500 MHz, DMSO-d_6_) δ 1.48 (3H, s, Me), 2.35 (2H, AB-system, Δ_AB_ = 0.03 ppm, *J*_AB_ = 14.2 Hz, H-7), 3.28 (1H, ddd, *J* = 11.2, 8.1, 6.1 Hz, H-3B), 3.75 (1H, ddd, *J* = 11.2, 8.2, 5.4 Hz, H-3A), 3.96 (1H, td, *J* = 8.2, 6.1 Hz, H-2B), 4.06 (1H, td, *J* = 8.2, 5.4 Hz, H-2A), 7.51 (1H, s, OH) ppm. ^13^C NMR (126 MHz, DMSO-d_6_) δ 24.2 (Me), 40.4 (C-3), 43.0 (C-7), 65.9 (C-2), 79.6 (q, *J* = 29.8 Hz, C-6), 95.4 (C-7a), 123.9 (q, *J* = 283.8 Hz, CF_3_), 169.6 (C-5) ppm. ^19^F NMR (470 MHz, DMSO-d_6_) δ 83.91 (s, CF_3_) ppm. IR ν 3347 (O–H), 2997, 2919 (C–H), 1699 (C=O), 1184–1108 (C–F) cm-1. HRMS (ESI): calcd. for C_8_H_11_F_3_NO_3_ [M + H]^+^ 226.0686; found 226.0682.

*(6R*,7aS*)-6-hydroxy-7a-methyl-6-(trifluoromethyl)tetrahydropyrrolo[2,1-b][1,3]oxazol-5(6H)-one (**4a^c^**)* (mixed with **5^ct^** (20%) and **4a^t^** (2%)). Yield 13% (243 mg, *method A*); white solid; m.p. 156–157°C (CHCl_3_–Et_2_O/1:1). ^1^H NMR (500 MHz, DMSO-d_6_) *δ* 1.38 (3H, s, Me), 2.29 (1H, dq, ^2^*J*_HH_ = 15.2, ^4^*J*_HF_ = 1.3 Hz, H-7B), 2.74 (1H, d, *J* = 15.2 Hz, H-7A), 3.30 (1H, ddd, *J* = 11.2, 8.2, 6.0, H-3B, overlapped with H_2_O), 3.85 (1H, ddd, *J* = 11.2, 8.2, 5.7 Hz, H-3A), 3.96 (1H, td, *J* = 8.2, 6.0 Hz, H-2B), 4.00 (1H, td, *J* = 8.2, 5.7 Hz, H-2A), 7.22 (1H, s, OH) ppm. ^19^F NMR (470 MHz, DMSO-d_6_) *δ* 83.63 (d, *J* = 1.3 Hz, CF_3_) ppm. IR ν 3332 (O–H), 2993, 2912 (C–H), 1703 (C=O), 1164–1091 (C–F) cm^−1^. HRMS (ESI): calcd. for C_8_H_11_F_3_NO_3_ [M + H]^+^ 226.0686; found 226.0689.

*(6R*,7aR*)-7a-ethyl-6-hydroxy-6-(trifluoromethyl)tetrahydropyrrolo[2,1-b][1,3]oxazol-5(6H)-one (****4b^t^****)* смсеь Yield 23% (495 mg, *method A*), 66% (789 mg, *method C*); white solid; m.p.134–136°C (CHCl_3_–Et_2_O / 1:1). ^1^H NMR (500 MHz, DMSO-d_6_) *δ* 0.90 (3H, t, *J* = 7.3 Hz, H-2’), 1.68 (1H, dq, *J* = 14.7, 7.3 Hz, H-1’B), 1.82 (1H, dq, *J* = 14.7, 7.3 Hz, H-1’A), 2.23 (1H, d, *J* = 14.5 Hz, H-7B), 2.42 (1H, d, *J* = 14.5 Hz, H-7A), 3.25 (1H, ddd, *J* = 11.2, 8.2, 6.4 Hz, H-3B), 3.78 (1H, ddd, *J* = 11.2, 7.9, 5.1 Hz, H-3A), 3.90–3.95 (1H, m, H-2B), 4.00 (1H, td, *J* = 8.2, 5.1 Hz, H-2A), 7.46 (1H, s, OH) ppm. ^13^C NMR (126 MHz, DMSO-d_6_) δ 7.8 (C-2’), 29.1 (C-1’), 40.1 (C-3), 41.1 (C-7), 65.6 (C-2), 79.1 (q, *J* = 29.8 Hz, C-6), 95.3 (C-7a), 123.9 (q, *J* = 284.4 Hz, CF_3_), 170.3 (C-5) ppm. ^19^F NMR (470.5 MHz, DMSO-d_6_) *δ* 83.89 (s, CF_3_) ppm. IR ν 3345 (O–H), 2983–2909 (C–H), 1701 (C=O), 1168–1149 (C–F) cm^−1^. HRMS (ESI): calcd. for C_9_H_13_F_3_NO_3_ [M + H]^+^ 240.0842; found 240.0840.

(*6R*,7aS*)-7a-ethyl-6-hydroxy-6-(trifluoromethyl)tetrahydropyrrolo[2,1-b][1,3]oxazol-5(6H)-one (**4b^c^**)*. (mixed with **4b^t^** in the ratio 90:10). Yield 18 % (387 mg, *method A*), 16% (191 mg, *method C*); white solid; m.p. 110–112°C (CHCl_3_–Et_2_O / 1:1). ^1^H NMR (500 MHz, DMSO-d_6_) δ 0.88 (3H, t, J = 7.3 Hz, H-2’), 1.47 (1H, dq, *J* = 14.7, 7.3 Hz, H-1’B), 1.80 (1H, dq, *J* = 14.7, 7.3 Hz, H-1’A), 2.17 (1H, d, *J* = 15.4 Hz, H-7B), 2.73 (1H, d, *J* = 15.4 Hz, H-7A), 3.28 (1H, ddd, *J* = 11.0, 7.8, 6.4 Hz, H-3B), 3.86 (1H, ddd, *J* = 11.0, 8.1, 5.4 Hz, H-3A), 3.90–3.97 (2H, m, H-2), 7.23 (1H, s, OH) ppm. ^13^C NMR (126 MHz, DMSO-d_6_) δ 7.7 (C-2’), 27.7 (C-1’), 39.7 (C-3, overlapped with DMSO), 41.5 (C-7), 64.3 (C-2), 78.5 (q, *J* = 30.3 Hz, C-6), 96.8 (C-7a), 124.0 (q, *J* = 285.1 Hz, CF_3_), 171.9 (C-5) ppm. ^19^F NMR (376 MHz, DMSO-d_6_) δ 83.79 (s, CF_3_) ppm. IR ν 3398 (O–H), 3007–2883 (C–H), 1710 (C=O), 1179–1093 (C–F) cm^−1^. HRMS (ESI): calcd. for C_9_H_13_F_3_NO_3_ [M + Na]^+^ 262.0661; found 262.0659.

*(6R*,7aR*)-7a-Butyl-6-hydroxy-6-(trifluoromethyl)tetrahydropyrrolo[2,1-b][1,3]oxazol-5(6H)-one (**4c^t^**) (mixed with **4c^c^** in the ratio 90:10).* Yield 39 % (938 mg, *method A*); white solid; m.p. 90–94°C. (CHCl_3_–Et_2_O / 1:1). ^1^H NMR (500 MHz, DMSO-d_6_) δ 0.89 (t, *J* = 7.0 Hz, 3H, H-4’), 1.28–1.35 (4H, m, H-3’, H-2’), 1.62–1.68 (1H, m, H-1’B), 1.76–1.83 (1H, m, H-1’A), 2.24 (1H, d, *J* = 14.5 Hz, H-7B), 2.42 (1H, d, *J* = 14.5 Hz, H-7A), 3.25 (1H, ddd, *J* = 11.2, 8.0, 6.4 Hz, H-3B), 3.77 (1H, ddd, *J* = 11.2, 7.9, 5.2 Hz, H-3A), 3.91 (1H, td, *J* = 8.2, 6.4 Hz, H-2B), 4.00 (1H, td, *J* = 8.2, 5.2 Hz, H-2A), 7.46 (1H, s, OH) ppm. ^13^C NMR (126 MHz, DMSO-d_6_) δ 13.9 (C-4’), 22.2 (C-3’), 25.3 (C-2’), 35.9 (C-1’), 40.5 (C-3), 41.0 (C-7), 65.6 (C-2), 79.1 (q, *J* = 29.8 Hz, C-6), 97.5 (C-7a), 124.0 (q, *J* = 284.3 Hz, CF_3_), 170.2 (C-5) ppm. ^19^F NMR (470 MHz, DMSO-d_6_) δ 83.87 (s, CF_3_) ppm. IR ν 3324 (O–H), 2996–2878 (C–H), 1710 (C=O), 1179–1148 (C–F) cm^−1^. HRMS (ESI): calcd. for C_11_H_17_F_3_NO_3_ [M + H]^+^ 268.1155; found 268.1156.

*(6R*,7aS*)-7a-Butyl-6-hydroxy-6-(trifluoromethyl)tetrahydropyrrolo[2,1-b][1,3]oxazol-5(6H)-one (**4c^c^**).* Yield 26 % (625 mg, *method A*); white solid; m.p. 129–130 °C (Et_2_O). ^1^H NMR (400 MHz, DMSO-d_6_) δ 0.88 (3H, t, *J* = 7.0 Hz, H-4’), 1.27–1.33 (4H, m, H-3’, H-2’), 1.43 (1H, dq, *J* = 14.7, 7.0 Hz, H-1’B), 1.79 (1H, dq, *J* = 14.7, 7.0 Hz, H-1’A), 2.19 (1H, d, *J* = 15.3 Hz, H-7B), 2.73 (1H, d, *J* = 15.3 Hz, H-7A), 3.28 (1H, ddd, *J* = 11.0, 7.5, 6.9 Hz, H-3B), 3.85 (1H, ddd, *J* = 11.0, 7.9, 5.6 Hz, H-3A), 3.91–3.95 (2H, m, H-2), 7.22 (1H, s, OH) ppm. ^13^C NMR (126 MHz, DMSO-d_6_) δ 13.8 (C-4’), 22.1 (C-3’), 25.3 (C-2’), 34.5 (C-1’), 39.7 (C-3, overlapped with DMSO), 41.5 (C-7), 64.4 (C-2), 78.5 (q, *J* = 30.1 Hz, C-6), 96.5 (C-7a), 124.1 (q, *J* = 285.1 Hz, CF_3_), 171.9 (C-5) ppm. ^19^F NMR (376 MHz, DMSO-d_6_) δ 83.76 (s, CF_3_) ppm. IR ν 3409 (O–H), 3008–2866 (C–H), 1709 (C=O), 1178–1112 (C–F) cm^−1^. HRMS (ESI): calcd. for C_11_H_17_F_3_NO_3_ [M + H]^+^ 268.1155; found 268.1154.

*(6R*,7aR*)-6-Hydroxy-7a-phenyl-6-(trifluoromethyl)tetrahydropyrrolo[2,1-b][1,3]oxazol-5(6H)-one (**4d^c^**).* Yield 58 % (1499 mg, *method A*); white solid; m.p. 155–157°C (CHCl_3_–Et_2_O / 1:1). ^1^H NMR (500 MHz, DMSO-d_6_) δ 2.70 (1H, d, *J*_AB_ = 15.5 Hz, H-7B), 2.75 (1H, d, *J*_AB_ = 15.5 Hz, H-7A), 3.16 (1H, ddd, *J* = 11.2, 8.5, 4.6 Hz, H-3B), 3.51 (1H, td, *J* = 8.5, 6.7 Hz, H-2B), 3.91 (1H, ddd, *J* = 11.2, 8.3, 6.7 Hz, H-3A), 4.10 (1H, td, *J* = 8.5, 4.6 Hz, H-2A), 7.36–7.44 (6H, m, Ph, OH) ppm. ^13^C NMR (126 MHz, DMSO-d_6_) δ 42.3 (C-3), 43.2 (C-7), 64.6 (C-2), 78.5 (q, C-6, *J* = 30.6 Hz), 96.8 (C-7a), 123.9 (q, CF_3,_ *J* = 284.9 Hz), 124.9 (Co), 128.5 (Cp), 128.6 (Cm), 139.8 (Ci), 172.6 (C-5) ppm. ^19^F NMR (470 MHz, DMSO-d_6_) δ 83.65 (s, CF_3_). IR ν 3394 (O–H), 3076–2978 (C–H), 1713 (C=O), 1179–1149 (C–F) cm^−1^. C_13_H_12_F_3_NO_3_ (287.24). Calculated: C, 54.36; H, 4.21; N, 4.88; Found: C, 54.37; H, 4.22; N, 4.89. HRMS (ESI): calcd. for C_13_H_13_F_3_NO_3_ [M + H]^+^ 288.0842; found 288.0842.

*Ethyl (R*)-3,3,3-trifluoro-2-hydroxy-2-(((6R*,7aS*)-6-hydroxy-5-oxo-6-(trifluoromethyl)tetrahydropyrrolo[2,1-b]oxazol-7a(5H)-yl)methyl)propanoate (**5^tc^**).* Yield 47% (1671 mg, *method B*), 56% (1106 mg, *method D*); white solid; m.p. 168–170°C. (MeCN). ^1^H NMR (500 MHz, DMSO-d_6_) *δ* 1.25 (3H, t, *J* = 7.1 Hz, OCH_2_CH_3_), 2.20 (1H, d, *J* = 14.0 Hz, H-1’B), 2.29 (1H, d, *J* = 14.9 Hz, H-7”B), 2.63 (1H, br. d, *J* = 14.0 Hz, H-1’A), 2.99 (1H, d, *J* = 14.9 Hz, H-7”A), 3.35–3.41 (1H, m, H-3”B), 3.78–3.84 (2H, m, H-2”B, H-3”A), 3.86–3.92 (1H, m, H-2”A), 4.17 (1H, dq, *J* = 10.8, 7.1 Hz, OCH^B^CH_3_), 4.29 (1H, dq, *J* = 10.8, 7.1 Hz, OCH^A^CH_3_), 7.08 (1H, d, *J* = 1.6 Hz, C^2^-OH), 7.56 (1H, s, C^6”^-OH) ppm. ^13^C NMR (126 MHz, DMSO-d_6_) *δ* 13.7 (OCH_2_CH_3_), 37.9 (C-1’), 40.4 (C-3”), 42.1 (C-7”), 62.3 (OCH_2_CH_3_), 66.3 (C-2”), 76.0 (q, *J* = 27.5 Hz, C-2), 78.8 (q, *J* = 30.0 Hz, C-6”), 95.3 (C-7a”), 123.7 (q, *J* = 283.8 Hz, CF_3_), 123.9 (q, *J* = 288.6 Hz, CF_3_), 167.8 (C-1), 169.9 (C-5”) ppm. ^19^F NMR (470 MHz, DMSO-d_6_) *δ* 83.83 (s, 3F, CF_3_), 84.92 (s, 3F, CF_3_) ppm. IR ν 3456, 3284 (O–H), 2997–2924 (C–H), 1736, 1705 (C=O), 1167–1093 (C–F) cm^−1^. HRMS (ESI): calcd. for C_14_H_15_F_3_NO_3_ [M + H]^+^ 396.0876; found 396.0879.

*Ethyl (S^*^)-3,3,3-trifluoro-2-hydroxy-2-(((6R^*^,7aS)-6-hydroxy-5-oxo-6-(trifluoromethyl)tetrahydropyrrolo[2,1-b]oxazol-7a(5H)-yl)methyl)propanoate (**5^cc^**).* Yield 18% (640 mg, *method B*), 15% (296 mg, *method D*); white solid; m.p. 134–136 °C. (CHCl_3_–Et_2_O/2:1). ^1^H NMR (500 MHz, DMSO-d_6_) *δ* 1.23 (3H, t, *J* = 7.1 Hz, CH_3_), 2.26 (1H, d, *J* = 14.0 Hz, H-7”B), 2.50 (1H, d, *J* = 15.0 Hz, H-1’B), 2.61 (1H, d, *J* = 14.0 Hz, H-7”A), 2.67 (1H, dd, *J* = 15.0, 1.5 Hz, H-1’A), 3.40–3.46 (1H, m, H-3”B), 3.58–3.63 (m, 1H, H-2”B), 3.78–3.84 (m, 2H, H-2”A, H-3”A), 4.17 (dq, *J* =10.8, 7.1 Hz, 1H, OCH^B^), 4.24 (dq, *J* =10.8, 7.1 Hz, 1H, OCH^A^), 7.00 (d, *J* = 1.5 Hz, 1H, C^2^-OH), 7.56 (s, 1H, C^6”^-OH). ^13^C NMR (126 MHz, DMSO-d_6_) *δ* 13.6 (OCH_2_CH_3_), 40.8 (C-1’), 42.6 (C-3”), 44.6 (C-7”), 62.1 (OCH_2_), 66.7 (C-2”), 75.4 (q, *J* = 27.5 Hz, C-2), 78.6 (q, *J* = 29.8 Hz, C-6”), 95.4 (C-7a”), 123.7 (q, *J* = 284.2 Hz, CF_3_), 124.1 (q, *J* = 288.4 Hz, CF_3_), 167.8 (C-1), 170.2 (C-5”). ^19^F NMR (376 MHz, DMSO-d_6_) *δ* 84.23 (s, 3F, CF_3_), 85.64 (s, 3F, CF_3_). IR ν 3487, 3316 (O–H), 2991 (C–H), 1738, 1705 (C=O), 1169–1090 (C–F) cm^−1^. HRMS (ESI): calcd. for C_14_H_15_F_3_NO_3_ [M + H]^+^ 396.0876; found 396.0872.

*Ethyl 2-hydroxy-4-oxo-2-(trifluoromethyl)hexanoate (**6b**).* A mixture of 1530 mg (9 mmol) of ethyltrifluoropyruvate **1**, 522 mg (1.8 mmol) of 2-butanone **2b**, and 12 mg (0.1 mmol) of L-proline in 10 mL of DMF was placed in a flat-bottomed flask. The reaction mass was stirred for 2 days at a temperature (50℃). After completion of the reaction (TLC and ^19^F NMR monitoring), the reaction mixture was poured into water (100 mL) and the organic layer was extracted with chloroform (3 × 50 mL). The solvent was concentrated on a rotary evaporator. The residue was purified by column chromatography (eluent: CHCl_3_–hexane / 2:1). A yellow oil was isolated. Yield 1699 mг (78%). ^1^H NMR (500 MHz, DMSO-d_6_) *δ* 0.89 (3H, t, *J* = 7.1 Hz, CH_3_^Et^), 1.19 (3H, t, *J* = 7.1 Hz, CH_3_^OEt^), 3.07 (1H, d, *J* = 17.2 Hz, H-3A), 3.19 (1H, d, *J* = 17.2 Hz, H-3B), 2.44 (2H, dq, *J* = 11.0, 7.3 Hz, H-B^Et^), 2.53 (2H, dq, *J* = 11.0, 7.3 Hz, H-A^Et^, ), 4.13–4.23 (2H, m, CH_2_^OEt^), 6.75 (1H, d, *J* = 0.7 Hz, OH) ppm. ^13^C NMR (126 MHz, DMSO-d_6_) *δ* 7.2 (C-6), 13.67 (OCH_2_CH_3_), 35.7 (C-5), 44.4 (C-3), 61.8 (OCH_2_CH_3_), 74.9 (q, *J* = 28.1 Hz, C-2), 123.9 (q *J* = 287.2 Hz, CF_3_), 167.7 (C-1), 205.4 (C-4) ppm. ^19^F NMR (376 MHz, DMSO-d_6_) *δ* 84.83 (s, CF_3_) ppm. IR ν 3482 (O–H), 2987–2908 (C–H), 1747, 1626 (C=O), 1222–1097 (C–F) cm^−1^. HRMS (ESI): calcd. for C_9_H_14_F_3_O_4_ [M + H]^+^ 243.0839; found 243.0833.

*(7R*,8aR*)-7-Hydroxy-8a-methyl-7-(trifluoromethyl)tetrahydro-2H-pyrrolo[2,1-b][1,3]oxazine-6(7H)-one (**8a******^t^****) (mixed with **8a^c^** in the ratio 84:16).* Yield 28 % (603 mg, *method A*), 30 % (358 mg, *method C*); white solid; m.p. 96–98°C (CHCl_3_–Et_2_O / 1:1). ^1^H NMR (500 MHz, DMSO-d_6_) δ 1.47–1.58 (2H, m, H-3), 1.61 (3H, c, Me), 2.27 (2H, AB-system, Δ_AB_ = 0.06 ppm, *J*_AB_ = 14.4 Hz, H-8), 3.21 (1H, dm, *J* = 13.0 Hz, H-4B), 3.75 (1H, dm, *J* = 11.5 Hz, H-2B), 3.86 (1H, dm, *J* = 13.0 Hz, H-4A), 3.98 (1H, dm, *J* = 11.5 Hz, H-2A), 7.21 (1H, s, OH) ppm. ^13^C NMR (126 MHz, DMSO-d_6_) δ 20.3 (Me), 24.7 (C-3), 35.2 (C-4), 43.8 (C-8), 60.6 (C-2), 74.9 (q, *J* = 29.9 Hz, C-7), 86.2 (C-8a), 124.4 (q, *J* = 285.1 Hz, CF_3_), 166.2 (C-6), ppm. ^19^F NMR (470 MHz, DMSO-d_6_) δ 84.22 (s, CF_3_). IR ν 3430, 3351 (O–H), 2990–2885 (C–H), 1696 (C=O), 1168–1057 (C–F) cm^−1^. HRMS (ESI): calcd. for C_9_H_13_F_3_NO_3_ [M + H]^+^ 240.0842; found 240.0849.

*(7R*,8aS*)-7-Hydroxy-8a-methyl-7-(trifluoromethyl)tetrahydro-2H-pyrrolo[2,1-b][1,3]oxazine-6(7H)-one (**8a^c^**) (mixed with **8a^t^** in the ratio 89:11 ).* Yield 44% (947 mg, *method A*), 62% (741 mg, *method C*); white solid; m.p. 76–78 °C (CHCl_3_–Et_2_O/1:1). ^1^H NMR (500 MHz, DMSO-d_6_) δ 1.49–1.64 (5H, m, Me, H-3), 2.15 (1H, br.d, *J* = 15.2 Hz, H-8B), 2.57 (1H, d, *J* = 15.2 Hz, H-8A), 3.24 (1H, td, *J* = 13.0, 4.0 Hz, H-4B), 3.76 (1H, dm, *J* = 12.2 Hz, H-2B), 3.88 (1H, dm, *J* = 13.0 Hz, H-4A), 4.01 (1H, td, *J* = 12.2, 2.9 Hz, H-2A), 7.13 (1H, s, OH) ppm. ^13^C NMR (126 MHz, DMSO-d_6_) δ 20.3 (Me), 24.7 (C-3), 35.2 (C-4), 43.8 (C-8), 60.6 (C-2), 74.9 (q, *J* = 29.9 Hz, C-7), 86.2 (C-8a), 124.4 (q, *J* = 285.1 Hz, CF_3_), 166.2 (C-6) ppm. ^19^F NMR (470 MHz, DMSO-d_6_) δ 83.71 (s, CF_3_). IR ν 3392, 3296 (O–H), 2997–2887 (C–H), 1702 (C=O), 1175–1058 (C–F) cm^−1^. HRMS (ESI): calcd. for C_9_H_13_F_3_NO_3_ [M + H]^+^ 240.0842; found 240.0850.

*(7R*,8aR*)-8a-Ethyl-7-hydroxy-7-(trifluoromethyl)tetrahydro-2H-pyrrolo[2,1-b][1,3]oxazine-6(7H)-one (**8b^t^**).* Yield 21 % (478 mg, *method A*), 24 % (304 mg, *method C*); white solid; m.p. 68–70°C (CHCl_3_–Et_2_O / 1:1). ^1^H NMR (500 MHz, DMSO-d_6_) *δ* 0.84 (3H, t, *J* = 7.3 Hz, H-2’), 1.49–1.55 (2H, m, H-3), 1.68 (1H, dq, *J* = 14.5, 7.3 Hz, H-1’B), 2.10 (1H, d, *J* = 14.8 Hz, H-8B), 2.28–2.36 (2H, m, H-8A, H-1’A), 3.16–3.22 (1H, m, H-4B), 3.72 (1H, dm, *J* = 12.1 Hz, H-2B), 3.84–3.90 (2H, m, H-4A, H-2A), 7.17 (1H, s, OH) ppm. ^13^C NMR (126 MHz, DMSO-d_6_) *δ* 7.1 (C-2’), 24.29 and 24.30 (C-3, C-1’), 35.0 (C-4), 41.0 (C-8), 60.0 (C-2), 74.9 (q, *J* = 30.0 Hz, C-7), 88.5 (C-8a), 124.3 (q, *J* = 284.6 Hz, CF_3_), 165.5 (C-6) ppm. ^19^F NMR (376 MHz, DMSO-d_6_) *δ* 84.14 (s, CF_3_) ppm. IR (ATR) ν 3240 (O–H), 2970–2889 (C–H), 1674 (C=O), 1161–1092 (C–F) cm^−1^. HRMS (ESI): calcd. for C_10_H_15_F_3_NO_3_ [M + H]^+^ 254.0999; found 254.1006.

*(7R*,8aS*)-8a-Ethyl-7-hydroxy-7-(trifluoromethyl)tetrahydro-2H-pyrrolo[2,1-b][1,3]oxazine-6(7H)-one (**8b******^c^****) (mixed with **8b^t^** in the ratio 81:19).* Yield 27 % (615 mg, *method A*), 62 % (784 mg, *method C*); white solid; m.p. 98–100°C (CHCl_3_–Et_2_O / 1:1). ^1^H NMR (500 MHz, DMSO-d_6_) *δ* 0.82 (3H,t, *J* = 7.3 Hz, H-2’), 1.47–1.71 (3H, m, H-3, H-1’B), 2.00 (1H, d, *J* = 15.3 Hz, H-8B), 2.38 (1H, dq, *J* = 14.6, 7.3 Hz, H-1’A), 2.51 (1H, d, *J* = 15.3 Hz, H-8A, overlapped with DMSO), 3.22 (1H, td, *J* = 13.0, 3.8 Hz, H-4B), 3.73 (1H, dm, *J* = 12.2 Hz, H-2B), 3.87 (1H, dm, *J* = 13.0 Hz, H-4A), 3.92 (1H, td, *J* = 12.2, 2.7 Hz, H-2A), 7.13 (1H, s, OH) ppm. ^13^C NMR (126 MHz, DMSO-d_6_) *δ* 6.9 (C-2’), 23.9 and 24.3 (C-1’, C-3), 35.5 (C-4), 40.4 (C-8), 60.1 (C-2), 74.8 (q, *J* = 29.8 Hz, C-7), 88.6 (C-8a), 124.4 (q, *J* = 285.0 Hz, CF_3_), 167.0 (C-6) ppm. ^19^F NMR (470 MHz, DMSO-d_6_) *δ* 83.85 (s, CF_3_) ppm. IR (ATR) ν 3347 (O–H), 2983–2895 (C–H), 1695 (C=O), 1196–1131 (C–F) cm^−1^. HRMS (ESI): calcd. for C_10_H_15_F_3_NO_3_ [M + H]^+^ 254.0999; found 254.1019.

*(7R*,8aR*)-8a-Butyl-7-hydroxy-7-(trifluoromethyl)tetrahydro-2H-pyrrolo[2,1-b][1,3]oxazin-6(7H)-one (**8c^t^**) (mixed with **8c^c^** in the ratio 90:10).* Yield 24 % (608 mg, *method A*); white solid; m.p. 80–83°C (CHCl_3_–Et_2_O / 1:1). ^1^H NMR (500 MHz, DMSO-d_6_) *δ* 0.91 (3H, t, *J* = 7.3 Hz, H-4’), 1.19–1.37 (4H, m, H-2’, H-3’), 1.48–1.54 (2H, m, H-3), 1.63 (1H, ddd, *J* = 14.2, 10.8, 5.2 Hz, H-1’B), 2.12 (1H, d, *J* = 14.8 Hz, H-8B), 2.31 (1H, ddd, *J* = 14.2, 10.7, 5.7 Hz, H-1’A), 2.34 (1H, d, *J* = 14.8 Hz, H-8A), 3.18–3.24 (1H, m, H-4B), 3.72 (1H, dm, *J* = 12.0 Hz, H-2B), 3.83–3.91 (2H, m, H-2A, H-4A), 7.16 (1H, s, OH) ppm. ^13^C NMR (126 MHz, DMSO-d_6_) *δ* 13.9 (C-4’), 22.0 (C-3’), 24.3 (C-3), 24.6 (C-2’), 31.1 (C-1’), 35.0 (C-4), 41.5 (C-8), 60.0 (C-2), 74.9 (q, *J* = 29.9 Hz, C-7), 88.2 (C-8a), 124.3 (q, *J* = 284.5 Hz, CF_3_), 165.4 (C-6) ppm.^19^F NMR (376 MHz, DMSO-d_6_) *δ* 84.13 (s, CF_3_) ppm. IR ν 3305 (O–H), 2974–2869 (C–H), 1682 (C=O), 1174–1150 (C–F) cm^−1^. HRMS (ESI): calcd. for C_12_H_19_F_3_NO_3_ [M − H]^–^ 280.1166; found 280.1167.

*(7R*,8aS*)-8a-Butyl-7-hydroxy-7-(trifluoromethyl)tetrahydro-2H-pyrrolo[2,1-b][1,3]oxazin-6(7H)-one (**8c^c^**).* Yield 46 % (1164 mg, *method A*); white solid; m.p. 158–160°C (MeCN). ^1^H NMR (500 MHz, DMSO-d_6_) *δ* 0.90 (3H, t, *J* = 7.3 Hz, H-4’), 1.18–1.26 (2H, m, H-2’), 1.29–1.38 (2H, m, H-3’), 1.41–1.51 (2H, m, H-1’B, H-3B), 1.59 (1H, qt, *J* = 12.8, 5.2 Hz, H-3A), 2.02 (1H, d, *J* = 15.5 Hz, H-8B), 2.35–2.42‘ (1H, m, H-1’A), 2.51 (1H, d, *J* = 15.5 Hz, H-8A, overlapped with DMSO), 3.23 (1H, td, *J* = 13.2, 3.8 Hz, H-4B), 3.73 (1H, br. dd, *J* = 12.2, 5.2 Hz, H-2B), 3.87 (1H, br. dd, *J* = 13.2, 5.2 Hz, H-4A), 3.93 (1H, td, *J* = 12.2, 2.7 Hz, H-2A), 7.12 (1H, s, OH) ppm. ^13^C NMR (126 MHz, DMSO-d_6_) *δ* 13.9 (C-4’), 22.0 (C-3’), 24.3 (C-3), 24.6 (C-2’), 30.8 (C-1’), 35.5 (C-4), 41.0 (C-8), 60.2 (C-2), 74.8 (q, *J* = 29.9 Hz, C-7), 88.3 (C-8a), 124.4 (q_,_ *J* = 284.6 Hz, CF_3_), 166.9 (C-6). ^19^F NMR (470 MHz, DMSO-d_6_) *δ* 83.83 (CF_3_) ppm. IR ν 3313 (O–H), 2961–2869 (C–H), 1686 (C=O), 1189–1116 (C–F) cm^−1^. HRMS (ESI): calcd. for C_12_H_19_F_3_NO_3_ [M + H]^+^ 282.1312; found 282.1313.

*(7R*,8aR*)-7-Hydroxy-8a-phenyl-7-(trifluoromethyl)tetrahydro-2H-pyrrolo[2,1-b][1,3]oxazin-6(7H)-one (**8d^t^**).* Yield 58 % (1571 mg, *method A*); white solid; m.p. 143–145°C (MeCN). ^1^H NMR (500 MHz, DMSO-d_6_) *δ* 1.46 (1H, dm, *J* = 13.0 Hz, H-3B), 1.63 (1H, qt, *J* = 13.0, 5.1 Hz, H-3A), 2.22 (1H, d, *J* = 14.8 Hz, H-8B), 2.64 (1H, d, *J* = 14.8 Hz, H-8A), 2.97 (1H, td, *J* = 13.1, 3.8 Hz, H-4B), 3.54 (1H, td, *J* = 12.1, 2.2, H-2B), 3.83 (1H, ddm, *J* = 12.2, 4.6 Hz, H-2A), 4.02 (1H, ddm, *J* = 13.1, 5.2 Hz, H-4A), 7.23 (1H, s, OH), 7.32 (2H, dd, *J* = 8.3, 1.4 Hz, H*o*), 7.40 (1H, tt, *J* = 7.4, 1.4 Hz, H*p*), 7.49 (2H, t, *J* = 7.6 Hz, H*m*) ppm. ^13^C NMR (126 MHz, DMSO-d_6_) *δ* 24.2 (C-3), 36.4 (C-4), 46.2 (C-8), 61.9 (C-2), 75.0 (q, *J* = 30.0 Hz, C-7), 89.8 (C-8a), 124.1 (q *J* = 284.9 Hz, CF_3_,), 125.8 (C*o*), 128.4 (C*p*), 129.2 (C*m*), 139.6 (C*i*), 167.1 (C-6) ppm. ^19^F NMR (470 MHz, DMSO-d_6_) *δ* 84.31 (s, CF_3_) ppm. IR ν 3290 (O–H), 2968–2876 (C–H), 1695 (C=O), 1199–1121 (C–F) cm^−1^. HRMS (ESI): calcd. for C_14_H_15_F_3_NO_3_ [M + H]^+^ 302.099; found 302.0997.

*Ethyl 3,3,3-trifluoro-2-hydroxy-2-[(7-hydroxy-6-oxo-7-(trifluoromethyl)tetrahydro-2H-pyrrolo[2,1-b][1,3]oxazin-8a(6H)-yl)methyl)propanoate (mixture of **9**:**9’** in the ratio ≈ 1:1).* Yield 69% (2540 mg, *method B*); white solid; m.p. 135–137°C. (CHCl_3_–Et_2_O / 4:1). ^1^H NMR (500 MHz, DMSO-d_6_) *δ* 1.20 (1.5H, t, *J* = 7.1 Hz, CH_3_^Et^), 1.25 (1.5H, t, *J* = 7.1 Hz, CH_3_^Et^), 1.47–1.59 (2H, m, H-3”), 1.69 (0.5H, d, *J* = 14.8 Hz), 2.02 (0.5H, d, *J* = 15.3 Hz), 2.06 (d, *J* = 15.7 Hz, 0.5H), 2.57 (0.5H, d, *J* = 15.2 Hz), 2.67 (0.5H, d, *J* = 15.2 Hz), 3.15 (0.5H, d, *J* = 15.3 Hz), 3.23 (0.5H, ddd, *J* = 13.4, 12.6, 3.7 Hz), 3.29–3.37 (1H, m, overlapped with H_2_O), 3.40 (d, *J* = 14.8 Hz, 0.5H), 3.65 (0.5H, dm, *J* = 12.5 Hz,), 3.73 (0.5H, dm, *J* = 12.5 Hz,), 3.80–3.90 (1.5H, m), 4.05–4.13 (1.5H, m), 4.18 (0.5H, dq, *J* = 10.8, 7.1 Hz, OCH^Et^), 4.23 (0.5H, dq, *J* = 10.8, 7.1 Hz, OCH^Et^), 7.10 (0.5H, s, OH), 7.12 (0.5H, s, OH), 7.16 (1H, s, OH) ppm. ^13^C NMR (126 MHz, DMSO-d_6_) *δ* 13.4 and 13.6 (CH_3_^Et^), 23.8 and 24.0 (C-3”), 33.3 and 33.5 (C-1’), 35.7 and 36.3 (C-8”), 39.7 and 40.7 (C-4”), 60.8 and 61.0 (C-2”), 62.2 and 62.4 (OCH_2_), 74.35 (q, *J* = 30.0, C-2), 74.38 (q, *J* = 30.0, C-2), 75.94 (q, *J* = 28.2, C-7”), 75.95 (q, *J* = 27.8, C-7”), 86.9 and 87.1 (C-8a “), 123.9 (q, *J* = 288.3, CF_3_), 124.0 (q, *J* = 288.3, CF_3_), 124.2 (q, *J* = 284.6, CF_3_), 124.3 (q, *J* = 285.0, CF_3_), 167.3, 167.4, 167.5 and 168.6 (C-1, C-6”) ppm. ^19^F NMR (470 MHz, DMSO-d_6_) *δ* 83.78 (s, 3F, CF_3_), 84.25 (s, 3F, CF_3_), 85.23 (s, 3F, CF_3_), 85.72 (s, 3F, CF_3_) ppm. IR ν 3482, 3315 (O–H), 2994–2903 (C–H), 1749, 1699 (C=O), 1173–1157 (C–F) cm^−1^. HRMS (ESI): calcd. for C_14_H_16_F_6_NO_6_ [M − H]^-^ 408.0887; found 408.0890.

*Ethyl 3,3,3-trifluoro-2-hydroxy-2-[(6-hydroxy-8a-methyl-5-oxo-6-(trifluoromethyl)tetrahydropyrrolo[2,1-b]oxazol-7a(5H)-yl)methyl]propanoate (mixture of **10**:**10’** in the ratio 56:44).* Yield 64% (2356 mg, *method B*); white solid; m.p. 133–134°C. (CHCl_3_–Et_2_O / 2:1). ^1^H NMR (500 MHz, DMSO-d_6_) δ 0.88 (d, *J* = 6.8 Hz, 1.3H, Me), 1.08 (d, *J* = 6.9 Hz, 1.7H, Me), 1.23 (t, *J* = 7.1 Hz, 3H, CH_3_^Et^), 2.27–2.35 (m, 1H, H-1’, H-7”), 2.62–2.73 (m, 2H, H-1’, H-7”), 3.40–3.62 (m, 2H, H-2”, H-3”), 3.76–3.84 (m, 2H, H-2”, H-3”), 4.16–4.23 (m, 2H, OCH_2_), 6.94, (d, *J* = 1.5 Hz, 0.44H, OH), 6.96 (d, *J* = 1.5 Hz, 0.56H, OH), 7.58 (s, 0.44H, OH), 7.66 (s, 0.56H, OH) ppm. ^13^C NMR (126 MHz, DMSO-d_6_) δ 7.7 and 8.4 (Me), 13.7 (CH_3_^Et^), 36.5 (C-1’), 41.6 and 42.9 (C-3”), 48.6 and 51.1 (C-7”), 62.0 (OCH_2_), 66.9 and 68.1 (C-2”), 75.2 (q, J = 27.1, C-2), 75.3 (q, *J* = 27.2, C-2), 79.2 (q, *J* = 28.3, C-6”), 80.7 (q, *J* = 28.3, C-6”), 97.3 and 97.5 (C-7a”), 123.6 (q, *J* = 285.2, CF_3_), 123.9 (q, *J* = 285.5, CF_3_), 124.2 (q, *J* = 288.7, CF_3_), 124.3 (q, *J* = 288.8, CF_3_), 167.9 and 168.0 (C-1), 169.2 and 169.4 (C-5”) ppm. ^19^F NMR (376 MHz, DMSO-d_6_) δ 85.39 (s, 1.7F, CF_3_), 85.60 (s, 1.3F, CF_3_), 86.40 (s, 1.7F, CF_3_), 89.73 (s, 1.3F, CF_3_) ppm. IR ν 3486, 3321 (O–H), 2992–2904 (C–H), 1736, 1702 (C=O), 1194–1114 (C–F) cm^−1^. HRMS (ESI): calcd. for C_14_H_18_F_6_NO_6_ [M + H]^+^ 410.1036; found 410.1033.

*Ethyl (R*)-3,3,3-trifluoro-2-hydroxy-2-(((R*)-4-hydroxy-1-(2-hydroxyethyl)-3-methyl-5-oxo-4-(trifluoromethyl)-4,5-dihydro-1H-pyrrol-2-yl)methyl)propanoate (**11a**).* Yield 8% (294 mg, *method B*); white solid; m.p. 141–143 °C. (CHCl_3_–Et_2_O / 2:1). ^1^H NMR (500 MHz, DMSO-d_6_) *δ* 1.22 (3H, t, *J* = 7.1 Hz, CH_3_), 1.62 (3H, s, C^3’^-Me), 3.17 (2H, s, H-1’), 3.40 (1H, ddt, *J* = 10.6, 8.5 5.0 Hz, H-2”‘B), 3.47 (1H, dq, *J* = 10.6, 4.7 Hz, H-2”‘A), 3.57 (1H, dt, *J* = 14.6, 4.7 Hz, H-1”‘B), 3.67 (1H, ddd, *J* = 14.6, 8.5, 5.0 Hz, H-1”‘A), 4.12 (1H, dq, *J* = 10.8, 7.1 Hz, OC-H^B^), 4.23 (1H, dq, *J* = 10.8, 7.1 Hz, OC-H^A^), 4.92 (1H, t, *J* = 4.9 Hz, C^2’”^-OH), 7.17 (1H, s, C^4’’^-OH), 7.26 (1H, s, C^2^-OH). ^13^C NMR (126 MHz, DMSO-d_6_) *δ* 8.7 (Me), 13.5 (CH_3_), 27.2 (CH_2_), 42.6 (C-1”‘), 58.8 (C-2’”), 62.5 (OCH_2_), 76.2 (q, *J* = 29.4, C-4’’), 77.4 (q, *J* = 27.4, C-2), 112.4 (C-3’’), 123.4 (q, *J* = 286.8, CF_3_), 123.9 (q, *J* = 288.1, CF_3_), 137.9 (C-2’’), 167.2 (C-1), 172.2 (C-5’’). ^19^F NMR (376 MHz, DMSO-d_6_) *δ* 85.67 (s, 3F, CF_3_), 85.69 (s, 3F, CF_3_). IR ν 3494, 3302 (O–H), 2996–2903 (C–H), 1744, 1670 (C=O), 1174–1132 (C–F) cm^−1^. HRMS (ESI): calcd. for C_14_H_18_F_6_NO_6_ [M + H]^+^ 410.1033; found 410.1034.

*Ethyl (S*)-3,3,3-trifluoro-2-hydroxy-2-(((R*)-4-hydroxy-1-(2-hydroxyethyl)-3-methyl-5-oxo-4-(trifluoromethyl)-4,5-dihydro-1H-pyrrol-2-yl)methyl)propanoate (**11b**).* Yield 47% (1360 mg, *method B*); white solid; m.p. 120–122°C. (Et_2_O–hexane / 3:1). ^1^H NMR (500 MHz, DMSO-d_6_) *δ* 1.22 (3H, t, *J* = 7.1 Hz, CH_3_), 1.55–1.61 (2H, m, H-2”‘), 1.63 (3H, s, C^3’’^-Me), 2.97 (1H, d, *J* = 15.0 Hz, H-1’B), 3.19 (1H, dd, *J* = 15.0, 1.5 Hz, H-1’A), 3.30–3.40 (2H,m, H-3”‘), 3.51 (1H, ddd, *J* = 14.5, 7.5, 6.8 Hz, H-1’’’B), 3.60 (1H, ddd, *J* = 14.5, 7.5, 7.0 Hz, H-1’’’A), 4.12 (1H, dq, *J* = 10.8, 7.1 Hz, OC-H^B^), 4.23 (1H, dq, *J* = 10.8, 7.1 Hz, OC-H^A^), 4.53 (1H, t, *J* = 4.9 Hz, C^3”‘^-OH), 7.14 (1H, s, C^4’’^-OH), 7.23 (1H, d, *J* = 1.5 Hz, C^2^-OH). ^13^C NMR (126 MHz, DMSO-d_6_) *δ* 8.73 (Me), 13.50 (CH_3_), 26.96 (CH_2_), 30.72 (C-2’”), 37.32 (C-1”‘), 57.73 (C-3”‘), 62.58 (OCH_2_), 76.17 (q, *J* = 29.5, C-4’’), 77.24 (q, *J* = 27.4, C-2), 113.03 (C-3’’), 123.35 (q, *J* = 286.7, CF_3_), 123.83 (q, *J* = 288.1, CF_3_), 137.38 (C-2’’), 167.21 (C-1), 172.12 (C-5’’). ^19^F NMR (376 MHz, DMSO-d_6_) *δ* 85.64 (s, 3F, CF_3_), 85.74 (s, 3F, CF_3_) ppm. IR ν 3478, 3293 (O–H), 2992–2896 (C–H), 1751, 1670 (C=O), 1174–1134 (C–F) cm^−1^. HRMS (ESI): calcd. for C_15_H_20_F_6_NO_6_ [M + H]^+^ 424.1189; found 424.1186.

*Ethyl (R*)-3,3,3-trifluoro-2-hydroxy-2-(((R*)-4-hydroxy-1-(3-hydroxypropyl)-3-methyl-5-oxo-4-(trifluoromethyl)-4,5-dihydro-1H-pyrrol-2-yl)methyl)propanoate (**11b’**) (mixture of **11b**:**11b’** in the ratio 72:28).* Yield 76% (2893 mg, *method B*); white solid; m.p. 120–122 °C. (CHCl_3_–Et_2_O / 2:1). ^1^H NMR (500 MHz, DMSO-d_6_) *δ* 1.66 (3H, s, C^3’’^-Me), 3.08 (2H, AB-system, ∆_AB_ = 0.03 ppm, *J*_AB_ = 15.4 Hz, CH_2_), 7.18 (1H, s, C^4’’^-OH), 7.24 (1H, s, C^2^-OH) ppm; the signals of other protons coincide with the signals of the major diastereomer **11b**. ^13^C NMR (126 MHz, DMSO-d_6_) *δ* 8.86 (Me), 13.54 (CH_3_), 27.14 (CH_2_), 30.73 (C-2”‘), 37.18 (C-1’”), 57.57 (C-3’”), 62.58 (OCH_2_), 76.22 (q, *J* = 29.2, C-4’’), 77.02 (q, *J* = 28.0, C-2), 113.52 (C-3’’), 123.52 (q, *J* = 286.2, CF_3_), 123.83 (q, *J* = 288.1, CF_3_), 137.12 (C-2’’), 167.40 (C-1), 171.98 (C-5’’) ppm. ^19^F NMR (376 MHz, DMSO-d_6_) *δ* 85.81 (s, 3F, CF_3_), 86.01 (s, 3F, CF_3_) ppm. IR ν 3475, 3292 (O–H), 2992–2896 (C–H), 1752, 1716 (C=O), 1173–1135 (C–F) cm^−1^. HRMS (ESI): calcd. for C_15_H_20_F_6_NO_6_ [M + H]^+^ 424.1189; found 424.1184

### 3.5. XRD Experiments

The X-ray studies for compounds **4d^c^, 5^tc^, 8c^c^** were performed on an Xcalibur 3 CCD (Oxford Diffraction Ltd., Abingdon, UK) diffractometer with a graphite monochromator, λ(MoKα) 0.71073 Å radiation, T 295(2), for compound **11b** was registered on an XtaLAB Synergy (Oxford Diffraction Ltd., Abingdon, UK) diffractometer with hybrid pixel monochromator, λ(MoKα) 0.71073 Å radiation, T 295(2). An empirical absorption correction was applied. Using Olex2 [38], the structure was solved with the Superflip [39] structure solution program using charge flipping and refined with the ShelXL [40] refinement package using least squares minimization. All non-hydrogen atoms were refined in the anisotropic approximation; H-atoms at the C-H bonds were refined in the “rider” model with dependent displacement parameters. An empirical absorption correction was carried out through spherical harmonics, implemented in the SCALE3 ABSPACK scaling algorithm by a program “CrysAlisPro” (Rigaku Oxford Diffraction).

The full set of X-ray structural data for compounds **4d^c^, 5^tc^, 8c^c^, 11b** was deposited at the Cambridge Crystallographic Data Center (deposits CCDC-238736 (**4d^c^**), -2238737 (**5^tc^**), -2238738 (**8c^c^**)**,** -2238739 (**11b**).

*Crystal Data for **4d^c^***: C_13_H_12_F_3_NO_3_ (M = 287.24g/mol): triclinic, space group P-1, a = 6.0290(7) Å, b = 8.1542(10) Å, c = 13.4778(17) Å, α = 79.481(10)°, β = 86.443(10)°, γ = 75.029(10), V = 629.27(14) Å^3^, Z = 2, T = 295(2) K, μ(CuKα) = 0.136 mm^−1^, D_calc_ = 1.516 g/cm^3^, 5481 reflections measured to (7.37 ° ≤ 2Θ ≤ 61.83°), 3339 unique (R_int_ = 0.0434, R_sigma_ = 0.0881) which were used in all calculations. The final R_1_ was 0.0702 (I > 2σ(I)) and wR_2_ was 0.1634 (all data) (Appendix A, Appendix A).

*Crystal Data for **5^tc^**:* C_13_H_15_F_6_NO_6_ (M = 395.26 g/mol): monoclinic, space group P2_1_/n, a = 10.8844(8) Å, b = 13.1919(9) Å, c = 11.3734(9) Å, α = γ =90°, β = 92.641(7)°, V = 1631.3(2) Å^3^, Z = 4, T = 295(2) K, μ(CuKα) = 0.167 mm^−1^, D_calc_ = 1.609 g/cm^3^, 11283 reflections measured to (7.14° ≤ 2Θ ≤ 62.03°), 4404 unique (R_int_ = 0.0496, R_sigma_ = 0.0604) which were used in all calculations. The final R_1_ was 0.0544 (I > 2σ(I)) and wR_2_ was 0.1455 (all data) (Appendix A, Appendix A).

*Crystal Data for **8c^c^***: C_12_H_18_F_3_NO_3_ (M = 281.27 g/mol): monoclinic, space group P2_1_/c, a = 10.6332(12) Å, b = 13.8923(13) Å, c = 9.5664(15) Å, α = γ =90°, β = 102.235(13)°, V = 1381.0(3) Å^3^, Z = 4, T = 295(2) K, μ(CuKα) = 0.122 mm^−1^, D_calc_ = 1.353 g/cm^3^, 10666 reflections measured to (7.06° ≤ 2Θ ≤ 61.72°), 3779 unique (R_int_ = 0.0582, R_sigma_ = 0.0803) which were used in all calculations. The final R_1_ was 0.0611 (I > 2σ(I)) and wR_2_ was 0.1539 (all data) (Appendix A, Appendix A).

*Crystal Data for **11b**:* C_15_H_19_NO_6_F_6_ (M = 423.31g/mol): monoclinic, space group P21/c, a = 19.1431(10) Å, b = 6.4882(4) Å, c = 16.4430(9) Å, α = γ =90°, β = 107.551(6)°, V = 1947.2(2) Å^3^, Z = 4, T = 295(2) K, μ(CuKα) = 0.145 mm^−1^, D_calc_ = 1.444 g/cm^3^, 15118 reflections measured to (5.00° ≤ 2Θ ≤ 52.74°), 3972 unique (R_int_ = 0.1526, R_sigma_ = 0.0998) which were used in all calculations. The final R_1_ was 0.0601 (I > 2σ(I)) and wR_2_ was 0.1547 (all data) (Appendix A, Appendix A).

## 4. Conclusions

A method for the synthesis of bicyclic γ-lactam annulated oxazacycles has been developed based on the multicomponent reaction of ethyl trifluoropyruvate with methyl ketones and 1,2-, 1,3-amino alcohols. Thus, the use of aminoethanol makes it possible to obtain tetrahydropyrrolo[2,1-*b*]oxazol-5-ones, and the use of aminopropanol–tetrahydropyrrolo[2,1-*b*][1,3]oxazine-6-ones. Variation of the methyl ketone component creates opportunities for the introduction of various substituents at the nodal carbon atom of these bicycles. The method proposed by us is distinguished by the simplicity of execution and the availability of initial reagents.

It has been shown that the structure of final γ-lactams is determined by the stoichiometric amount of ethyl trifluoropyruvate and the nature of methyl ketone. At the same time, distinctive features of the transformations of amino alcohols were found in comparison with the previously studied reactions with 1,2-, 1,3-diamines, since the use of a double excess of ethyl trifluoropyruvate results in the formation of bicyclic aldols rather than tricyclic dipyrrolooxazacycles [35]. This feature is due to the fact that the oxygen atom in the cycle does not have the opportunity for subsequent addition reactions, and, consequently, the formation of tricycles. In addition, the reaction of a double excess of pyruvate with 2-butanone and aminopropanol stops at the stage of formation of N-hydroxypropyl-2,3-dihydropyrrol-5-one, the possibility of its isolation is probably due to the lower reactivity of the hydroxyl group compared to the amino function. It can also be noted that, in contrast to the transformations of diamines, three-component cyclizations with amino alcohols are less diastereoselective, since they predominantly lead to the formation of two *cis*- and *trans*-isomers, the diastereomeric structure of which we were able to reliably establish using NMR spectroscopy and X-ray diffraction.

The mechanism of formation of bicyclic γ-lactams has been determined. It represents successive domino reactions with the initial formation of an aldol and a bis-aldol from pyruvate and methyl ketone, which becomes possible due to the increased electrophilicity of the carbonyl group at the trifluoromethyl substituent.

The synthesized tetrahydropyrrolo[2,1-*b*]oxazol-5-ones and tetrahydropyrrolo[2,1-*b*][1,3]oxazine-6-ones are of interest both for biological testing and for the following chemical transformations, for example, oxazole ring opening reactions to obtain new γ-lactams.

## Data Availability

Not applicable.

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
