# Peer review of "Multicomponent Domino Cyclization of Ethyl Trifluoropyruvate with Methyl Ketones and Amino Alcohols as A New Way to γ-Lactam Annulated Oxazacycles"

_molecules, 2023, doi:10.3390/molecules28041983_

Round 1

Reviewer 1 Report

V. I. Saloutin and his group described the multi-component synthesis of bicyclic γ-lactams fused oxazole or oxazine ring by using methyl ketones, amino alcohols, and trifluoropyruvate.  They have succeeded in synthesizing highly valuable products using relatively inexpensive reagents.  Their results are well worthy of publication, although the selectivities and the yields of products were low to moderate.  The syntheses are well described and the structure of the obtained products are clearly established by NMR and X-Ray crystallographic analyses.

This manuscript was well organized, although there are a few weak points in this paper, especially in the stereoselectivity of the products.  So, I think the authors should clarify the following issues before publishing the manuscript on Molecule.

1.        In this paper, there is no tentative rationale to explain the results, in particular the syn/anti stereoselectivities of 4d and 8d were different.  Therefore, after clarification on the model of their cyclizations, I invite the authors to try to explain their results with the help of good schemes.

2.        The double aldol reaction did not proceed by using hexan-2-one but acetone and methyl ethyl ketone proceed.  Why not?  Steric hindrance?

3.        Why the reaction did not proceed via path b to give X5?  The authors should add the reason in the text.

Additional comments:

1)        Line 22 on page 1; ‘N’ should be shown in italics.

2)        The configuration of products is not cis/trans but syn/anti.  In addition, please check and correct all of the product numbers (e.g. 4ac to 4as) in the text and schemes.

3)        In Scheme 2; Isolated yields? If so, show it in the footnote.

4)        Line 167 on page 5; The number of ‘pyruvate 1’ should be shown in bold.

5)        In Scheme 6; 4a,bt, 8a,bc, 8a,bt? Did you mean ‘4at, bt, 8ac, bc, 8at, bt’ ?

6)        Page 8; Some of ‘trans-’ should be shown in italics.

7)        Line 249-250 on page 8; The sentence should be moved to an earlier page, which would be easier to understand for readers - e.g. page 3?

8)        Line 265 on page 9; ‘8cc’ should be shown in bold.

9)        Line 280 on page 9; The parenthesis ‘Figure 4, b)’ should be shown a normal font.

10)    In Scheme 8; ‘3’ should be shown in bold.

11)    The proposed mechanism is not a catalytic cycle.  Please rewrite the mechanism using straight arrows, and delete path b and the route from 4 to 5.

Author Response

We thank the Reviewer 1 for the detailed evaluation of our manuscript. We have improved the manuscript according to the reviewer's suggestions.

Reviewer’s original comments are listed below followed by our response (Author) to each comment.

  1. Reviewer: In this paper, there is no tentative rationale to explain the results, in particular the syn/anti stereoselectivities of 4dand 8d were different. Therefore, after clarification on the model of their cyclizations, I invite the authors to try to explain their results with the help of good schemes.

Author: We think that the use of acetophenone as a methyl ketone component does not change the three-component cyclization route proposed in Scheme 8. Most likely, the stereoselectivity of the formation of heterocycles 4dc and 8dt is associated with the introduction of a bulky phenyl substituent, which acts as a conformational anchor stabilizing the most favorable diastereomeric form. Obviously, the existence of heterocycles 4dc and 8dt as different diastereomers is determined by the structure of the neighboring oxazacycle. In the case of a less conformationally flexible oxazole ring, a cis - isomer is formed, while a more conformationally flexible oxazine ring favors the formation of a trans - isomer.

We have added to the text of the article a sentence about the stereoselective features of the formation of heterocycles 4dc and 8dt

  1. Reviewer: The double aldol reaction did not proceed by using hexan-2-one but acetone and methyl ethyl ketone proceed.  Why not?  Steric hindrance?

Author: The double aldol reaction does not occur in the case of hexan-2-one, since the nucleophilicity of its α-methylene center in the butyl substituent is significantly reduced under the influence of electronic and steric factors. The corresponding addition has been made to the article.

  1. Reviewer:Why the reaction did not proceed via path b to give X5?  The authors should add the reason in the text.

Author: According to the reviewer's note (see item 11), we removed the path b and the route from 4 to 5, since they are not implemented.

Additional comments:

1) Reviewer: Line 22 on page 1; ‘N’ should be shown in italics.

Author: Inaccuracy has been corrected.

2) Reviewer: The configuration of products is not cis/trans but syn/anti.  In addition, please check and correct all of the product numbers (e.g. 4acto 4as) in the text and schemes.

Author: IUPAC recommends using syn- and anti-nomenclature for compounds in which the single bond between the two centers is free to rotate, while the term cistrans stereoisomers is used for compounds with double bonds that do not rotate, or for ring structures , where the rotation of bonds is restricted or prevented [Basic Organic Stereochemistry, Chapter 3, Stereoisomers, E. L. Eliel, S. H. Wilen, M. P. Doyle., John Wiley & Sons, Inc., 2001; IUPAC "Gold Book" diastereoisomerism doi:10.1351/goldbook.D01679]. In this regard, we use cis, trans nomenclature for the cycles described in the work.

3) Reviewer: In Scheme 2; Isolated yields? If so, show it in the footnote.

Author: Scheme 2 shows the yields of isolated products. We indicated this in the title to the Scheme 2.

4) Reviewer: Line 167 on page 5; The number of ‘pyruvate 1’ should be shown in bold.

Author: Inaccuracy has been corrected.

5) Reviewer: In Scheme 6; 4a,bt8a,bc8a,bt? Did you mean ‘4atbt8acbc8atbt’ ?

Author: Inaccuracy has been corrected.

6) Reviewer: Page 8; Some of ‘trans-’ should be shown in italics.

Author: Inaccuracy has been corrected.

7) Reviewer: Line 249-250 on page 8; The sentence should be moved to an earlier page, which would be easier to understand for readers - e.g. page 3?

Author: We moved the sentence with information about cis- and trans-configuration to page 3, where it is first mentioned.

8) Reviewer: Line 265 on page 9; ‘8cc’ should be shown in bold.

Author: Inaccuracy has been corrected.

9) Reviewer: Line 280 on page 9; The parenthesis ‘Figure 4, b)’ should be shown a normal font.

Author: Inaccuracy has been corrected.

10) Reviewer: In Scheme 8; ‘3’ should be shown in bold.

Author: Inaccuracy has been corrected.

11) Reviewer: The proposed mechanism is not a catalytic cycle. Please rewrite the mechanism using straight arrows, and delete path b and the route from 4 to 5.

Author: The mechanism of the investigated three-component reaction is autocatalyzed, since amino alcohols play the role of a catalyst that starts the reaction of aldol addition and a reagent for creating heterocycles, so we use round arrows.

We have removed the path b and the route from 4 to 5 as they are not implemented.

Reviewer 2 Report

Victor I. Saloutin and co-workers described a method for the synthesis of fluorinated bicyclic γ-lactam annulated oxazacycles using Multicomponent domino cyclization of ethyltrifluoropyruvate with methyl ketones and amino alcohols.

The γ-lactams are important molecules as well as fluorinated compounds for their biological interest. Here the authors succeed, by using multicomponent reactions, to establish a mechanism of formation of the various interesting compounds obtained although the yields are limited for a use in therapy.

It is a quality work and difficult to carry out which can bring important information for the scientific community. This is why I recommend the publication of this article in Molecules but I have a few comments to take into account for a more fluid reading of the article.

It would be more understandable for the reader to indicate from the top of diagram 1 the chemical formulas of 5tc and 5cc and to give a legend for “tc” and “cc” which is not mentioned in the text either. We have to wait for line 140 to understand that “c” is used for the cis-isomer and “t” for the trans-isomer. line 167 in "pyruvate 1 the "1" should be bolded. At the level of the conditions mentioned in Scheme 6, in terms of solvents, it is mentioned DMF or 1,4-dioxane but why not to have tried the reaction with 2 components like that with three components in the THF to compare them at best? Could you explain this choice? line 265 8cc should be bolded.  

Author Response

Thanks to Reviewer 2 for reviewing our manuscript. We have improved the manuscript according to the reviewer's suggestions.

Reviewer’s original comments are listed below followed by our response (Author) to each comment.

1) Reviewer: It would be more understandable for the reader to indicate from the top of diagram 1 the chemical formulas of 5tcand 5cc and to give a legend for “tc” and “cc” which is not mentioned in the text either. We have to wait for line 140 to understand that “c” is used for the cis-isomer and “t” for the trans-isomer.

Author: We have added transcripts of cis- and trans-abbreviations to the text of the article on page 3.

2) Reviewer: line 167 in "pyruvate 1 the "1" should be bolded.

Author: Inaccuracy has been corrected.

3) Reviewer: At the level of the conditions mentioned in Scheme 6, in terms of solvents, it is mentioned DMF or 1,4-dioxane but why not to have tried the reaction with 2 components like that with three components in the THF to compare them at best? Could you explain this choice?

Author: Aldols 6a,b were obtained under conditions (DMF and proline as a catalyst), which we previously successfully used for the synthesis of aldol 6a [Goryaeva, M.V., Fefelova, O.A., Burgart, Y.V. et al. A Three-Component Synthesis of trifluoromethylated hexahydropyrrolo[1,2-a]imidazol-5-ones and hexahydropyrrolo[1,2-a]pyrimidin-6-ones. Chem. Heterocycl. Comp. 2022, 58, 421–431, https://doi.org/10.1007/s10593-022-03108-4].

Bicycles 4 and 8 were synthesized from aldols 6 in 1,4-dioxane similarly to three-component reactions, and they were isolated in good yields, so we began to perform additional optimization of the conditions.

4) Reviewer: line 265 8ccshould be bolded.  

Author: Inaccuracy has been corrected.